# A Systematic Review and Meta-Analysis of Cognitive Effects of rTMS in Caucasian Patients with Mild Cognitive Impairment

**DOI:** 10.3390/brainsci13091335

**Published:** 2023-09-17

**Authors:** Christiane Licht, Swetlana Herbrandt, Carmen van Meegen, Hartmut Lehfeld, Thomas Hillemacher, Kneginja Richter

**Affiliations:** 1University Clinic for Psychiatry and Psychotherapy, Paracelsus Medical University, 90419 Nuremberg, Germanykneginja.richter@pmu.ac.at (K.R.); 2Statistical Consulting and Analysis, Center for Higher Education, TU Dortmund University, 44227 Dortmund, Germany; 3CuraMed Tagesklinik GmbH, 90411 Nuremberg, Germany; 4Faculty for Social Sciences, Technical University for Applied Sciences Georg Simon Ohm, 90489 Nuremberg, Germany

**Keywords:** mild cognitive impairment, repetitive transcranial magnetic stimulation, meta-analysis, cognition, review

## Abstract

In recent years, repetitive transcranial magnetic stimulation (rTMS) has received much attention as a non-invasive, effective treatment modality for mild cognitive impairment (MCI). Although several meta-analyses have reported that rTMS can improve cognitive abilities, improvements in individual memory domains (speech, language, concentration, and memory) are poorly understood. In addition, stimulation parameters may be flawed in studies of global populations because of ethnic differences between Caucasians and Asians. This meta-analysis aimed to systematically characterize the efficacy of different combinations of rTMS parameters on different cognitive domains in Caucasian patients with MCI. We conducted a systematic literature search in Medline PubMed, Pubpsych, and Embase on the use of rTMS in MCI patients through November 2022. Randomized, double-blind, and sham-controlled trials (RCTs) from the Caucasian patient population were included. The studies reported outcome measures for different domains of cognition, such as language, concentration, or memory. Possible effects of covariates were examined using meta-regressions. The search yielded five publications. The analyses found that rTMS improved cognitive functions, memory, concentration, and language in patients with MCI and treatment with rTMS compared with the sham stimulation group. The statistical analysis results of the studies showed that rTMS could improve various cognitive functions, such as memory and concentration, in Caucasian MCI patients. A particular effect was found at a frequency of 10 Hz and stimulation of the LDLPFC. However, further studies are needed to validate these findings and explore more effective stimulation protocols and targets.

## 1. Introduction

Mild cognitive impairment (MCI) is an intermediate stage between normal aging and dementia [1,2]. The prevalence of MCI is between 16 and 20% for people over 65 years [1,2,3]. In about 20% of all patients, the mild impairment progresses to manifest dementia within one year [1,2,3]. Various studies show that currently used medications are ineffective in alleviating MCI symptoms [2,4]. Therefore, MCI is a prevalent, disabling, and challenging illness for which new treatment options are needed [2,4]. In recent years, repetitive transcranial magnetic stimulation (rTMS) has proven to be a promising new treatment approach. Current research suggests that alternating between high and low-frequency rTMS can significantly improve memory function over the long term. However, the efficacy of rTMS in patients with MCI has not been fully elucidated due to the low statistical power and heterogeneity of previous studies [1,2]. Several recent meta-analyses have examined various rTMS effects on cognition in patients with Alzheimer’s disease. The studies mainly included patients with prodromal Alzheimer’s disease, i.e., MCI or amnestic MCI (aMCI), and Alzheimer’s [5,6,7]. A recent meta-analysis reported on the effect of rTMS on cognition in patients with MCI, although the publication did not refer exclusively to randomized controlled trials (RCTs) [8]. From our knowledge, only one meta-analysis has exclusively investigated patients worldwide with MCI [9]. Although various studies have frequently reported that treatment with rTMS can improve cognition in MCI patients, the optimal stimulation protocols and application parameters are poorly understood [1,2,3,4,5,6,7,8,9]. Here, the rTMS frequency is one of the main factors affecting cortical activity [10]. However, there is no consensus on the optimal number of rTMS pulses required to achieve cortical excitability.

In addition, evidence points to inherent differences in cortical plasticity between Caucasians and Asians. Also, there is evidence that there appear to be differences between rTMS measures and, respectively, outcomes of different ethnic groups [11,12,13]. Likewise, studies are often so heterogeneous that various tests are used to assess overall cognition. This heterogeneity often hampers the assessment of performance in different cognitive domains. Although several meta-analyses have reported that rTMS can improve cognition in MCI patients [14,15,16], the improvements in specific cognitive domains are poorly understood. In addition, stimulation parameters in studies of unselected samples may be flawed because of ethnic differences between Caucasians and Asians. Due to the heterogeneity of the different studies, this meta-analysis is based on strict inclusion criteria depending on published RCTs in the European population. The pooled effects of rTMS were analyzed to assess the impact and safety of rTMS on depression and various cognitive functions in memory, concentration, speech, and language in Caucasian patients with MCI.

## 2. Materials and Methods

The present study conducted a systematic literature search on rTMS in MCI patients in November 2022. The study was registered in the internal study center of Paracelsus Medical University (registration number: FMS_FP_051.23-XI-1). The databases used were Medline PubMed, Pubpsych, and Embase, according to the Preferred Reporting Items for Systematic Reviews and Meta-Analyses (PRISMA) guidelines [17]. Following search terms for PubMed: (“MCI” OR “mild cognitive impairment” OR “cognition” OR “cogn”) AND (“rTMS” OR “transcranial magnet stimulation” OR “TBS” OR “theta burst stimulation”). An equivalent search was performed in Pubpsych using the following search terms: (“MCI” OR “mild cognitive impairment” OR “cognition” OR “cogn”) AND (“rTMS” OR “transcranial magnet stimulation” OR “TBS” OR “theta burst stimulation”). In Embase, we used the following search terms: (“MCI” OR “mild cognitive impairment” OR “cognition” OR “cogn”) AND (“rTMS” OR “transcranial magnet stimulation” OR “TBS” OR “theta burst stimulation”). Studies on humans were included if published between 1 January 2000 and 31 December 2022. All references found were downloaded into Zotero, and duplicates were deleted. The reference lists of the identified articles were screened for additional publications, which were added if applicable.

Inclusion Criteria: (1) randomized controlled studies investigating the effects of rTMS on cognitive function in Caucasian patients with MCI; (2) participants diagnosed with MCI based on any diagnostic criteria, such as the Mini-Mental Status Test or the Petersen criteria [18,19]; (3) the experimental group received rTMS treatment; (4) the control group received sham rTMS stimulation; and (5) outcomes included the global cognitive ability and specific cognitive domains as determined by neuropsychological tests in which a mean and a standard deviation or confidence interval was given for each test result.

Titles and abstracts, followed by full texts, were reviewed by authors, and hits were scored according to the PICO scheme and PRISMA checklist for studies [17,20]. Inconclusive judgments were resolved by consensus. Articles were classified for the first author, year of publication, number of participants, mean age, standard deviation, using cognitive tests, rTMS site, rTMS frequency, rTMS intensity, number of given pulses, and days of treatment. If test results were reported at different time points, the data were labeled for each time point: T1 = test results before any rTMS, T2 = test results immediately after finishing the whole rTMS session. Because the interval between cognitive testing differed in the various studies, additional testing time points were designated later as T3 and T4 (Table 1). To further investigate the factors influencing the effects of rTMS on overall cognitive outcomes, the following four subgroup analyses were conducted: effects on (1) depression, (2) memory, (3) concentration, and (4) speech and language.

The effect of rTMS on cognitive function in patients with MCI has been defined based on Hedges’ [21] standardized mean difference (SMD). Based on this, multivariate random-effect models were computed [22,23]. As the mean difference in the change in cognitive outcome measures depends on different influencing variables, we considered three SMD: (a) rTMS—control group (treatment), (b) final—baseline (time point), and (c) T2—baseline (time point). Given the various cognitive tests used in the included studies, a categorization according to (1)–(4) was used to summarize the pooled effect sizes of the eligible studies. All subtests described in different studies were distributed as follows: Beck’s depression inventory II, geriatric depression scale, and brief neuropsychological test battery = (1) depression. Rivermead behavioral memory test, logical memory test, story recall test, list recall test, rey auditory–verbal learning test, list-learning test, story memory test, figure-copy test, digit-span memory test, and mini-mental state test = (2) memory. Letter-number-sequencing test, trail-making test A and B, line orientation test, attentive matrices test, and Stroop effect test = (3) concentration. Verbal-fluency animal-naming test, picture-naming test, semantic fluency test, phonemic verbal fluency test, and semantic verbal fluency test = (4) speech and language. Other tests that could not be assigned to a category are described individually. For the calculation of the models, the direction of action of individual outcomes was adjusted so that all outcomes indicate a healthier status at higher values. This was necessary because not all tests indicate improvement at higher values. Depending on the measurement instrument, lower values can also show an improvement. Pre-, post-, and follow-up stimulus scale scores, if published, were quoted for each group. In addition, we performed funnel plots to evaluate publication bias. We used the software R (R Core Team R version 4.3.1) [24] and the meta-analysis package metafor [25] for all calculations. In this analysis, the *p*-value was considered significant at *p* < 0.05. Thus, the significance level was 0.05.

## 3. Results

### 3.1. Findings of the Literature Search

The number of publications found and selected during the process is indicated in Figure 1. We retrieved a total number of 1069 studies from the selected databases and removed 85 duplicates. In addition, 818 articles were excluded because MCI was not the topic of the studies. The remaining 166 were further assessed for eligibility. A total of 32 studies were excluded after reading their titles and abstracts because no RCT was conducted. In addition, we excluded 127 studies after full-text reading because MCI was not the main topic. Finally, seven studies [26,27,28,29,30,31,32] were included in the analysis. The mean values and standard deviations of the test results were missing in the two included studies. The meta-analysis was, therefore, performed based on the remaining five studies [26,27,28,29,32], fulfilling the criteria for data extraction and analysis (Figure 1). The baseline characteristics of the included studies are presented in Table 1.

Published data were available for five studies included in the present meta-analysis. In total, 60 participants were randomly assigned to the active rTMS group, while the number of participants randomly assigned to the sham rTMS group was 64. Only one study stimulated the right hemisphere [31]. The remaining studies used a stimulation above the left dorsolateral prefrontal cortex (LDLPFC). The used frequencies varied between 5 [25,28] and 10 Hz [26,27,31]. The stimulation intensity was between 80 and 120% of the resting motor threshold. The number of pulses ranged between 500 and 3000 from 1 to 30 days of treatment. Follow-up tests were performed between 0.5 and 190 days after the baseline measurements before the first rTMS (T1). Similar stimulation parameters were applied in the control group. The coils were placed vertically for sham stimulation, or a special sham stimulation coil was used instead.

### 3.2. Subgroup Analysis

#### 3.2.1. Comparison between the rTMS and Control Group

The results are shown in Table 2. There is no evidence of bias in the funnel plot generated (Figure 2). The meta-analyzed studies showed no significant improvement or worsening in depressive symptoms, concentration, or speech and language compared to the control group at any time points T1–T4/T3. In the subgroup memory, there was a tendency for significant improvement in the rTMS group compared with the control group at time point T2 (estimate = 0.237, 95% CI = [−0.059, 0.533]). In the further course, a significant improvement in the rTMS group at time T3 became apparent (estimate = 0.422, 95% CI = [0.065, 0.780]). At time point T4, there was a positive estimate, but it was no longer significant with a *p*-value of 0.178.

#### 3.2.2. Comparison between Different Time Points T1 to T4/T3 in the rTMS and Control Group

The results are shown in Table 3. There is no evidence of bias in the funnel plot generated (Figure 3). The meta-analyzed studies showed no significant improvement or worsening between the T1 and T4/T3 time points in depressive symptoms, concentration, or speech and language. However, an interesting result was obtained concerning apathy. Here, a significant improvement of the symptomatology could be achieved in comparison from time T1 to time T4/T3 in the rTMS group (estimate = 1.981, 95% CI = [0.064, 3.898]). Also, in memory, there was a significant improvement in the rTMS group compared to time points T1 to T4 (estimate = 0.596, 95% CI = [0.138, 1.055]). The meta-analyzed studies for the control group showed no significant improvement or worsening between the time points T1 and T4 in apathy, memory, or concentration. However, a significant improvement in the categories of depression (estimate = 0.723, 95% CI = [−0.045, 1.400]), speech, and language (estimate = 0.828, 95% CI = [0.133, 1.524]) could be observed in comparison from time T1 to time T4 in the control group.

#### 3.2.3. Other Effects on Memory and Concentration in the rTMS and Control Group

In addition, we analyzed the possible effects of rTMS site, rTMS frequency, rTMS intensity, and number of given pulses on the category’s memory and concentration in the rTMS group and control group of T1 compared with T4. In the rTMS group, there was a significant improvement in the memory parameters with treatment over the left hemisphere (estimate = 0.796, 95% CI = [0.373, 1.218]), but no significant improvement over the right hemisphere (estimate = −0.223, 95% CI = [−1.248, 0.802]). Similar results were analyzed in the control group (Table 4). In the rTMS group, when treated over the left hemisphere, there was no significant improvement or deterioration in concentration compared with treatment over the right hemisphere. Similar results were seen in the control group (Table 5). When considering the different frequencies, there was no significant improvement or deterioration in memory, either in the rTMS group or in the control group of T1 compared with T4 (Table 6).

In the rTMS group, there was a significant worsening in concentration between T1 and T4 at a frequency of 5 Hz (estimate = −0.897, 95% CI = [−1.782, −0.012]). There were no differences at other stimulation frequencies in the rTMS or the control group (Table 7). Considering the different stimulation intensities, memory was significantly improved when stimulated at and above 100% in the rTMS group compared to the control groups at timepoint T1 compared with T4 (Table 8).

The number of pulses impacted memory: The memory improved in the rTMS group after stimulation with 1500, 2000, and 3000 pulses compared with the control groups at timepoint T1 compared with T4 (Table 9).

Also, for the number of treatment days, the memory category improved after treatment longer than ten days at T1 compared to T4 in both the rTMS and the control groups (Table 10). In the according funnel plots, no significant publication bias was recognized.

## 4. Discussion

The conducted literature search has yielded a total of seven publications encompassing Caucasian patients diagnosed with MCI.

It is important to note that the foundation of this review rests upon a relatively modest base of seven papers.

For calculation basis, sufficient data sets were available for five publications with 60 Caucasian patients. The outcome indicators in the rTMS group were compared with those in the control group, who received only sham rTMS.

The analyses revealed that rTMS improved cognitive functioning, especially in the category of memory. This effect was more pronounced when applying rTMS over the left DLPFC. According to research evidence, it is known that the DLPFC is involved in regulating executive functions such as working memory and cognitive flexibility [33,34]. One possible explanation is that the left DLPFC, in particular, is connected to other regions, forming one of the essential areas in the central executive network. Evidence shows that by regulating brain networks, rTMS can improve working memory in patients [33]. Likewise, it is conceivable that stimulation of the DLPFC may improve emotional feelings in patients, indirectly allowing memory improvement. Probably, the same can be assumed for the category concentration [34,35].

In the present meta-analysis, no significant improvement in depressive symptoms was detected. However, this could be due to the heterogeneous test procedures and the small sample sizes in the included studies. Another reason may be the difficulty of distinguishing memory disorders in depressed patients (pseudodementia). Another bias could be associated with pharmacological therapy that could influence the cognition and the symptoms of depression. The diagnosis of depression should be made after a clinical interview, including the psychometric diagnosis. In studies included in our analysis, the depressive symptoms have been evaluated chiefly solely by questionnaires, which could also be a source of bias. Concentration and memory disturbances are present both in depressive disorders and in MCI.

The rTMS frequency is one of the main factors affecting cortical activity. However, studies suggest that the number of pulses may also influence the regulation of brain excitability [35,36,37]. However, there is no consensus on the optimal number of rTMS pulses required to achieve cortical excitability. According to our meta-analysis, memory improved after stimulation with pulse values of 1500 and above. Further studies should be conducted to determine the benefits of rTMS on cognition with different numbers of pulses in patients with MCI.

These effects may be due to the limited number of available studies, especially in the Caucasian population. Specifically, only five MCI studies were identified with Caucasians, compared to five studies analyzed in studies with Asians [9]. Second, the ceiling effect of the cognitive tests can potentially limit the ability to detect changes [37] in performance before and after rTMS, particularly in the MCI population. Lastly, not all studies examine MCI populations with precisely the same age and gender. A comparison of rTMS’s effect on global cognition between MCI patients and younger individuals with cognitive deficits of the same gender could be interesting. In line with this, the differential impact of rTMS on functional networks could affect organization and associative memory in young and older adults in the Caucasian population.

Also, a few limitations should be considered when interpreting our study’s results. Using different scales to measure global cognition across other papers likely contributes to the high heterogeneity. There were only a small number of studies on MCI patients. More rTMS studies, specifically in the MCI population, will be needed to confirm the efficacy of this method. Lastly, because our study focused on immediate outcomes, future studies will be required to investigate the long-term clinical utility of rTMS on various effects in the MCI population.

Further studies should be conducted to determine the benefits of rTMS on the emotion and cognition function, especially in patients with MCI. In the future, new research results could also address the question of individual differences between Caucasians and Asians regarding plasticity during brain stimulation. Although the effects of rTMS on cognition showed some sustained effects after treatment in the included articles, this meta-analysis cannot fully address the sustainability of the impact. Studies with larger sample sizes are required to determine the best stimulation targets for rTMS that yield optimal emotional and cognitive improvement.

## Figures and Tables

**Figure 1 brainsci-13-01335-f001:**
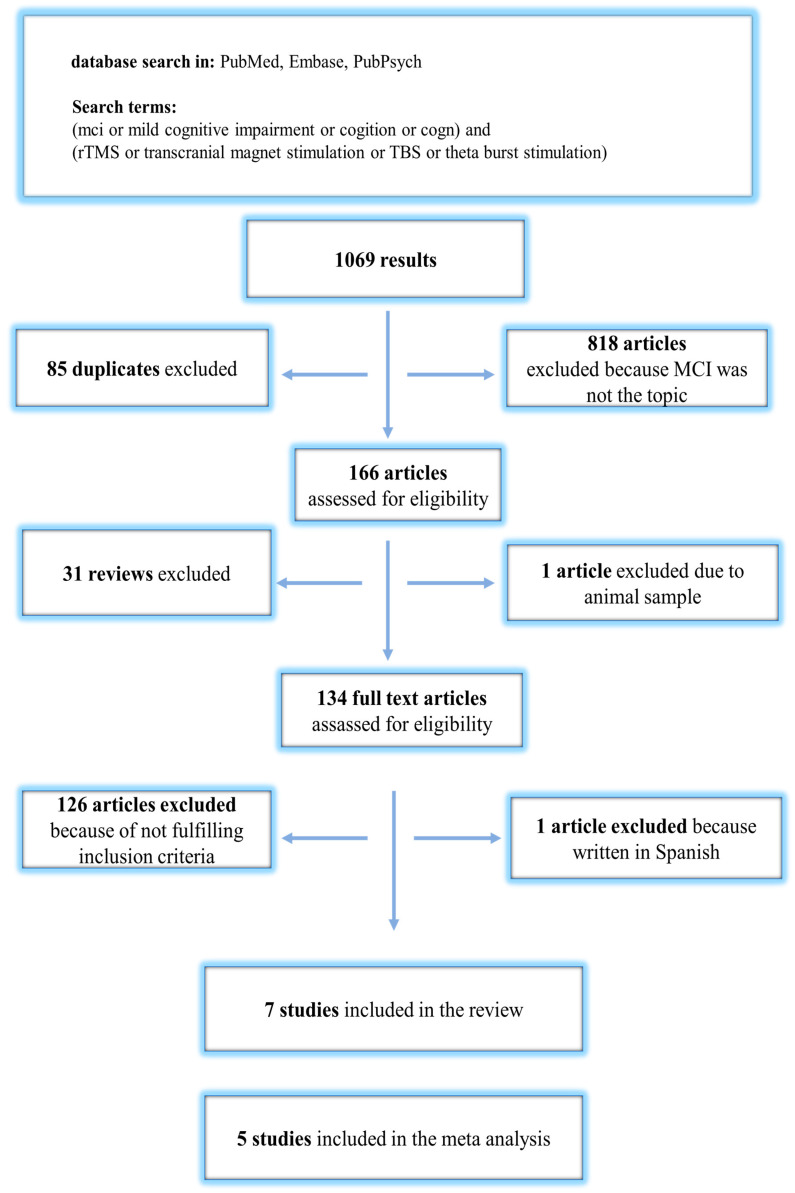
Flow chart of publication search and selection process.

**Figure 2 brainsci-13-01335-f002:**
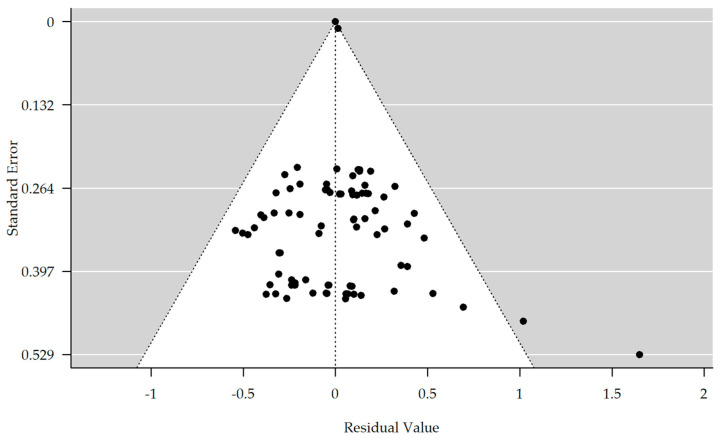
Funnel plot showing no significant publication bias (rank correlation test for funnel plot asymmetry: Kendall’s tau = 0.071, *p*-value = 0.346).

**Figure 3 brainsci-13-01335-f003:**
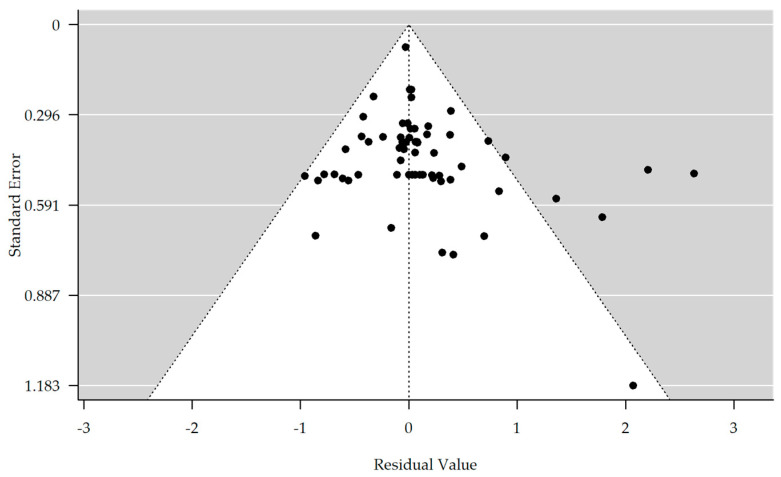
Funnel plot showing no significant publication bias (rank correlation test for funnel plot asymmetry: Kendall’s tau = 0.148, *p*-value = 0.085).

**Table 1 brainsci-13-01335-t001:** Description of the studies included in the final sample. The studies were chosen as specified in the method section. Abbreviations: SD = standard deviation, P = parallel study design, C = cross-over study design, LDLPFC = left dorsolateral prefrontal cortex, RIFG = right inferior frontal gyrus, # = no testing.

First Author	Year of Publication	Number of Patients	Mean Age	SD of the Age	Number of Controls	Age	SD	Study Design	rTMS Site	Frequency in Hz	Intensity in %	Number of Pulses	Days of Pulses	T2 in Days after T1	T3 in Days after T1	T4 in Days after T1
Marra	2015	15	65.10	3.50	19	65.20	4.10	P	LDLPFC	10	110	2000	10	10	30	#
Sole-Padulles	2006	20	66.95	9.43	19	68.68	7.78	P	LDLPFC	5	80	500	1	0.50	#	#
Padala	2018	4	68.00	10.00	5	64.00	9.00	C	LDLPFC	10	120	3000	10	10	30	50
Eliasova	2014	10	75.00	7.50	10	75.00	7.50	C	RIFG	10	90	2250	1	0.50	#	#
Roque Roque	2021	11	66.10	5.50	11	67.20	4.80	P	LDLPFC	5	100	1500	30	50	70	190

**Table 2 brainsci-13-01335-t002:** Comparison between the rTMS and control group in the categories (1)–(4) at any time point T1–T4. SE = standard error, 95% CI = 95% confidence interval.

	Estimate	SE	Z-Value	*p*-Value	95% CI	
T1 and category = depression	−0.021	0.306	−0.070	0.945	−0.622	0.579
T2 and category = depression	−0.673	0.442	−1.523	0.128	−1.539	0.193
T3 and category = depression	−0.081	0.427	−0.188	0.851	−0.918	0.757
T4 and category = depression	−0.140	0.304	−0.462	0.644	−0.736	0.455
T1 and category = memory	0.083	0.151	0.555	0.579	−0.211	0.378
T2 and category = memory	0.237	0.151	1.567	0.117	−0.059	0.533
T3 and category = memory	0.422	0.182	2.315	0.021	0.065	0.780
T4 and category = memory	0.319	0.237	1.345	0.178	−0.146	0.783
T1 and category = concentration	0.118	0.154	0.770	0.441	−0.183	0.419
T2 and category = concentration	0.060	0.154	0.392	0.695	−0.241	0.362
T3 and category = concentration	0.278	0.221	1.260	0.208	−0.155	0.711
T4 and category = concentration	0.162	0.432	0.376	0.707	−0.684	1.008
T1 and category = speech and language	0.597	0.355	1.682	0.093	−0.099	1.292
T2 and category = speech and language	0.068	0.346	0.197	0.844	−0.610	0.747
T3 and category = speech and language	−0.048	0.346	−0.139	0.889	−0.727	0.630

**Table 3 brainsci-13-01335-t003:** Comparison between time points T1 to T4/T3 for the rTMS group and control group. SE = standard error, 95% CI = 95% confidence interval.

	Estimate	SE	Z-Value	*p*-Value	95% CI	
category = apathy and group = rTMS	1.981	0.978	2.025	0.043	0.064	3.898
category = depression and group = rTMS	0.577	0.335	1.722	0.085	−0.080	1.233
category = memory and group = rTMS	0.596	0.234	2.548	0.011	0.138	1.055
category = concentration and group = rTMS	0.055	0.242	0.227	0.821	−0.419	0.529
category = speech and language and group = rTMS	0.217	0.384	0.565	0.572	−0.536	0.970
category = apathy and group = control	0.409	0.656	0.624	0.533	−0.877	1.695
category = depression and group = control	0.723	0.346	2.091	0.037	0.045	1.400
category = memory and group = control	0.266	0.226	1.176	0.239	−0.177	0.710
category = concentration and group = control	−0.123	0.237	−0.517	0.605	−0.587	0.342
category = speech and language and group = control	0.828	0.355	2.334	0.020	0.133	1.524

**Table 4 brainsci-13-01335-t004:** Comparison between time points for the category memory and stimulation site from T1 to T4/T3 for the rTMS and control groups.

	Estimate	SE	Z-Value	*p*-Value	95% CI	
rTMS_site = LDLPFC and group = rTMS	0.796	0.216	3.689	2.250 × 10^−4^	0.373	1.218
rTMS_site = RIFG and group = rTMS	−0.223	0.523	−0.427	0.669	−1.248	0.802
rTMS_site = LDLPFC and group = control	0.415	0.205	2.018	0.044	0.012	0.817
rTMS_site = RIFG and group = control	−0.176	0.522	−0.336	0.737	−1.200	0.848

SE = standard error, 95% CI confidence interval.

**Table 5 brainsci-13-01335-t005:** Comparison between time points for the category concentration and stimulation site from T1 to T4/T3 for the rTMS and control groups.

	Estimate	SE	Z-Value	*p*-Value	95% CI	
rTMS_site = LDLPFC and group = rTMS	−0.068	0.376	−0.181	0.856	−0.805	0.668
rTMS_site = RIFG and group = rTMS	0.193	0.564	0.342	0.733	−0.913	1.298
rTMS_site = LDLPFC and group = control	−0.159	0.367	−0.433	0.665	−0.879	0.560
rTMS_site = RIFG and group = control	−0.083	0.564	−0.147	0.883	−1.189	1.023

SE = standard error, 95% CI confidence interval.

**Table 6 brainsci-13-01335-t006:** Comparison between time points for the category memory and frequency from T1 to T4/T3 for the rTMS and control groups.

	Estimate	SE	Z-Value	*p*-Value	95% CI	
rTMS frequency [Hz] = 5 and group = rTMS	0.739	0.380	1.944	0.052	−0.006	1.484
rTMS frequency [Hz] = 10 and group = rTMS	0.439	0.375	1.170	0.242	−0.296	1.174
rTMS frequency [Hz] = 5 and group = control	0.289	0.378	0.765	0.444	−0.452	1.030
rTMS frequency [Hz] = 10 and group = control	0.258	0.355	0.727	0.467	−0.437	0.953

SE = standard error, 95% CI confidence interval.

**Table 7 brainsci-13-01335-t007:** Comparison between time points for the category concentration and frequency from T1 to T4/T3 for the rTMS and control groups.

	Estimate	SE	Z-Value	*p*-Value	95% CI	
rTMS frequency [Hz] = 5 and group = rTMS	−0.897	0.451	−1.986	0.047	−1.782	−0.012
rTMS frequency [Hz] = 10 and group = rTMS	0.217	0.152	1.434	0.152	−0.080	0.514
rTMS frequency [Hz] = 5 and group = control	−0.787	0.446	−1.765	0.078	−1.661	0.087
rTMS frequency [Hz] = 10 and group = control	0.007	0.144	0.051	0.959	−0.274	0.289

SE = standard error, 95% CI confidence interval.

**Table 8 brainsci-13-01335-t008:** Comparison between time points for the category memory and intensity from T1 to T4/T3 for the rTMS and control groups.

	Estimate	SE	Z-Value	*p*-Value	95% CI	
rTMS intensity [%] = 80 and group = rTMS	0.529	0.392	1.351	0.177	−0.238	1.296
rTMS intensity [%] = 90 and group = rTMS	−0.223	0.501	−0.446	0.656	−1.205	0.759
rTMS intensity [%] = 100 and group = rTMS	0.981	0.326	3.006	0.003	0.341	1.621
rTMS intensity [%] = 110 and group = rTMS	0.804	0.351	2.292	0.022	0.116	1.491
rTMS intensity [%] = 120 and group = rTMS	2.685	1.179	2.279	0.023	0.375	4.995
rTMS intensity [%] = 80 and group = control	−0.203	0.394	−0.515	0.607	−0.975	0.570
rTMS intensity [%] = 90 and group = control	−0.176	0.500	−0.351	0.726	−1.156	0.805
rTMS intensity [%] = 100 and group = control	0.672	0.317	2.119	0.034	0.050	1.294
rTMS intensity [%] = 110 and group = control	0.760	0.328	2.314	0.021	0.116	1.404
rTMS intensity [%] = 120 and group = control	−0.480	0.684	−0.701	0.483	−1.820	0.861

SE = standard error, 95% CI confidence interval.

**Table 9 brainsci-13-01335-t009:** Comparison between time points for the category memory and number of pulses from T1 to T4/T3 for the rTMS and control groups.

	Estimate	SE	Z-Value	*p*-Value	95% CI	
No. of pulses = 500 and group = rTMS	0.529	0.392	1.351	0.177	−0.238	1.296
No. of pulses = 1500 and group = rTMS	0.981	0.326	3.006	0.003	0.341	1.621
No. of pulses = 2000 and group = rTMS	0.804	0.351	2.292	0.022	0.116	1.491
No. of pulses = 2250 and group = rTMS	−0.223	0.501	−0.446	0.656	−1.205	0.759
No. of pulses = 3000 and group = rTMS	2.685	1.179	2.279	0.023	0.375	4.995
No. of pulses = 500 and group = control	−0.203	0.394	−0.515	0.607	−0.975	0.570
No. of pulses = 1500 and group = control	0.672	0.317	2.119	0.034	0.050	1.294
No. of pulses = 2000 and group = control	0.760	0.328	2.314	0.021	0.116	1.404
No. of pulses = 2250 and group = control	−0.176	0.500	−0.351	0.726	−1.156	0.805
No. of pulses = 3000 and group = control	−0.480	0.684	−0.701	0.483	−1.820	0.861

SE = standard error, 95% CI confidence interval.

**Table 10 brainsci-13-01335-t010:** Comparison between time points for the category memory and days of pulses from T1 to T4/T3 for the rTMS and control groups.

	Estimate	SE	Z-Value	*p*-Value	95% CI	
days of pulses = 1 & group = rTMS	0.273	0.262	1.043	0.297	−0.240	0.786
days of pulses = 10 & group = rTMS	0.517	0.258	2.003	0.045	0.011	1.023
days of pulses = 30 & group = rTMS	0.705	0.225	3.139	0.002	0.265	1.146
days of pulses = 1 & group = control	−0.193	0.263	−0.735	0.463	−0.710	0.323
days of pulses = 10 & group = control	0.406	0.220	1.842	0.065	−0.026	0.837
days of pulses = 30 & group = control	0.092	0.219	0.419	0.675	−0.337	0.520

## Data Availability

The data that support the findings of this study are available from the corresponding author upon reasonable request.

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
