# Peer review of "A Systematic Review and Meta-Analysis of Cognitive Effects of rTMS in Caucasian Patients with Mild Cognitive Impairment"

_brainsci, 2023, doi:10.3390/brainsci13091335_

Round 1

Reviewer 1 Report

I have identified several concerns that, in my academic judgment, necessitate addressing.

These concerns pertain to:

I would like to suggest the authors to approach a review paper with series academic attitude. 

·      It is noteworthy that evidence indicates intrinsic disparities in cortical plasticity between Caucasians and Asians. Additionally, there exists evidence suggesting distinctions between rTMS measures and the corresponding outcomes within diverse ethnic groups [10–12]. However, the relevance of References 10-12 to the current topic and its alignment within the field of brain stimulation is tenuous at best. Moreover, the nature of this subject underscores the necessity for its treatment with utmost academic integrity and scientific rigor.

·      The conducted literature search has yielded a total of seven publications encompassing Caucasian patients diagnosed with Mild Cognitive Impairment (MCI). It is important to note that the foundation of this review rests upon a relatively modest base of seven papers. This paucity indicates that the topic is still in its formative stages and is marked by a considerable degree of uncertainty that potentially undermines the formulation of robust hypotheses.

·      The prevalence of MCI is between 16 and 20% for people over 65 years. I would suggest adding a ref.

·      Various studies show that currently used medications are ineffective in alleviating MCI symptoms. I would suggest adding a references

·      Although various studies have frequently reported that treatment with rTMS can improve cognition in MCI patients, the optimal stimulation protocols and application parameters are poorly understood. I would suggest adding a refeferences

·      Make sure you are reporting congruent references

·      Why this review is important, why this topic is important ? I would suggest improving the introduction.

·      The present study conducted a systematic literature search on rTMS in MCI patients  in November 2022. Why the last access to the literature is 22th, November. I would suggest updating the literature review.

·      Please address abbreviation/acronymic where they are used for the first time.

·      Finally, seven studies [25–31] were included in the analysis. The meta-analysis was therefore performed based on the remaining five studies [25–28, 31] fulfilling the criteria for data extraction and analysis (s. Figure 1).  Why do we need a review paper if there are only 5 literatures about this topic? In one day, the reader will be able to read all the available sources about the argument. 

·      Where the data shows any difference between Caucasian and Asian in terms of plasticity during brain stimulation?

Minor editing of English language required

Author Response

Comments and Suggestions for Authors

I have identified several concerns that, in my academic judgment, necessitate addressing.

These concerns pertain to:

I would like to suggest the authors to approach a review paper with series academic attitude. 

  • It is noteworthy that evidence indicates intrinsic disparities in cortical plasticity between Caucasians and Asians. Additionally, there exists evidence suggesting distinctions between rTMS measures and the corresponding outcomes within diverse ethnic groups [10–12]. However, the relevance of References 10-12 to the current topic and its alignment within the field of brain stimulation is tenuous at best. Moreover, the nature of this subject underscores the necessity for its treatment with utmost academic integrity and scientific rigor.
  • The conducted literature search has yielded a total of seven publications encompassing Caucasian patients diagnosed with Mild Cognitive Impairment (MCI). It is important to note that the foundation of this review rests upon a relatively modest base of seven papers. This paucity indicates that the topic is still in its formative stages and is marked by a considerable degree of uncertainty that potentially undermines the formulation of robust hypotheses.

Thank you for your constructive and detailed feedback. The Discussion was improved as suggested.

  • The prevalence of MCI is between 16 and 20% for people over 65 years. I would suggest adding a ref.

Thank you for your constructive and detailed feedback. References were added as suggested.

  • Various studies show that currently used medications are ineffective in alleviating MCI symptoms. I would suggest adding a references

Thank you for your constructive and detailed feedback. References were added as suggested.

  • Although various studies have frequently reported that treatment with rTMS can improve cognition in MCI patients, the optimal stimulation protocols and application parameters are poorly understood. I would suggest adding a refeferences

Thank you for your constructive and detailed feedback. References were added as suggested.

  • Make sure you are reporting congruent references
  • Why this review is important, why this topic is important? I would suggest improving the introduction.

Thank you for your constructive and detailed feedback. The introduction was improved.

  • The present study conducted a systematic literature search on rTMS in MCI patients in November 2022. Why the last access to the literature is 22th, November. I would suggest updating the literature review.

Thank you for your constructive and detailed feedback. Unfortunately, it is not possible to do the review again. We will discuss a new work for next year.

  • Please address abbreviation/acronymic where they are used for the first time.

Thank you for your constructive and detailed feedback. The text was corrected as suggested.

  • Finally, seven studies [25–31] were included in the analysis. The meta-analysis was therefore performed based on the remaining five studies [25–28, 31] fulfilling the criteria for data extraction and analysis (s. Figure 1).  Why do we need a review paper if there are only 5 literatures about this topic? In one day, the reader will be able to read all the available sources about the argument. 

We understand the criticism. Due to the few studies found, meta-analysis seems complicated. However, each of the analyzed studies contains various tests of cognition. To overview this abundance within one study and to work out a trend as a reader seems questionable. Therefore, the meta-analysis was deliberately designed of summarize the different studies for better effect size estimation.

  • Where the data shows any difference between Caucasian and Asian in terms of plasticity during brain stimulation?

Thank you for your constructive and detailed feedback. We have taken up this exciting question in our discussion. 

Comments on the Quality of English Language

Minor editing of English language required

Thank you for your feedback, we will use the language service and correct the whole manuscript as soon as possible.

Submission Date

27 July 2023

Date of this review

06 Aug 2023 14:37:58

Reviewer 2 Report

This meta-analysis aimed to systematically characterize the efficacy of different combinations of rTMS parameters on different cognitive 18
domains in Caucasian patients with MCI. The authors conducted a systematic literature search in Medline PubMed, Pubpsych, and Embase on the use of rTMS in MCI patients through November 2022. Randomized, double-blind, sham-controlled trials (RCTs) from the Caucasian patient population were included. The studies reported outcome measures for different domains of cognition, such as language, concentration, or memory. Possible effects of covariates were examined using meta-regressions. The search yielded five publications. The analyses found that rTMS improved cognitive functions, memory, concentration, and language in patients with MCI and treatment with rTMS compared with the sham stimulation group.

The manuscript deals with a topic of great interest and is well structured. I have only a few small suggestions for authors.

In the introduction, the authors might consider the following recent work investigating the effects of rTMS and motor coordination:

_ Moscatelli et al., High frequencies repetitive transcranial magnetic stimulation (rTMS) increase motor coordination performances in volleyball players, BMC Neuroscience, 2023, 24(1), 30.

"Flowchart 1" is blurry. If possible increase the resolution of the figure.

A "conclusions" paragraph should be inserted.

Author Response

Comments and Suggestions for Authors

This meta-analysis aimed to systematically characterize the efficacy of different combinations of rTMS parameters on different cognitive 18
domains in Caucasian patients with MCI. The authors conducted a systematic literature search in Medline PubMed, Pubpsych, and Embase on the use of rTMS in MCI patients through November 2022. Randomized, double-blind, sham-controlled trials (RCTs) from the Caucasian patient population were included. The studies reported outcome measures for different domains of cognition, such as language, concentration, or memory. Possible effects of covariates were examined using meta-regressions. The search yielded five publications. The analyses found that rTMS improved cognitive functions, memory, concentration, and language in patients with MCI and treatment with rTMS compared with the sham stimulation group.

The manuscript deals with a topic of great interest and is well structured. I have only a few small suggestions for authors.

In the introduction, the authors might consider the following recent work investigating the effects of rTMS and motor coordination:

_ Moscatelli et al., High frequencies repetitive transcranial magnetic stimulation (rTMS) increase motor coordination performances in volleyball players, BMC Neuroscience, 2023, 24(1), 30.

Thank you for your feedback. We have added your suggested article.

"Flowchart 1" is blurry. If possible increase the resolution of the figure.

Thank you for your feedback. We have improved the image resolution.

A "conclusions" paragraph should be inserted.

Thank you for your feedback. A “conclusion” paragraph is undesired from the journal’s guidelines, I guess. Maybe we can ask for an exception.

Submission Date

27 July 2023

Date of this review

03 Aug 2023 09:36:34

Reviewer 3 Report

The presented manuscript concerns a very important and current problem for aging societies. The manuscript meets the requirements for this type of work. It is stylistically correct, the introduction section contains all relevant information as an introduction to the work, the results are elaborated thoroughly and exhaustively. I'm just suggesting to improve the discussion section. In my opinion, it should be extended especially for information on how to use the presented results at the moment.

Author Response

Comments and Suggestions for Authors

The presented manuscript concerns a very important and current problem for aging societies. The manuscript meets the requirements for this type of work. It is stylistically correct, the introduction section contains all relevant information as an introduction to the work, the results are elaborated thoroughly and exhaustively. I'm just suggesting to improve the discussion section. In my opinion, it should be extended especially for information on how to use the presented results at the moment.

Thank you for your constructive and detailed feedback. The Discussion was improved as suggested.

Submission Date

27 July 2023

Date of this review

04 Aug 2023 12:41:40

Reviewer 4 Report

This is a systematic review and meta-analysis of cognitive effects of rTMS in Caucasian patients with mild cognitive impairment.

There are few comments here:

Page 2, line 69. The present study conducted a systematic literature search on rTMS in MCI patients in November 2022. The databases used were Medline PubMed, Pubpsych, and Embase, according….. The authors did not do the electronic search in Web of Science (WOS) database, which include the WOS Core Collection, Current Contents Connect, Derwent Innovations Index, KCI Korean Journal Database, Medline, Russian Science Citation Index, and SciELO Citation Index, which covers wider area. The other bigger database is Scopus. Scopus is larger than Web of Science and has more than 23,000 indexed journals in all scientific fields as studied by Falagas ME, Pitsouni EI, Malietzis GA, Pappas G. Comparison of PubMed, Scopus, Web of science, and Google scholar: strengths and weaknesses. FASEB J. 2008;22(2):338–42). If the authors searched in the WOS or Scopus, the outcome might be different (not only 7 studies as mentioned in page 3, line 139).

Page 2, line 84 to line 93. The exclusion criteria are the negative of the inclusion criteria. Redundant.

Page 2, line 93. Titles and abstracts, followed by full texts, were reviewed by authors, and hits were scored according to… Should start in new paragraph as the sentences are not part of Exclusion Criteria.

Page 3, line 142. Why (s. Figure 1)? Why not (Figure 1)?

Page 4, Figure 1. 126 articles were excluded because of not fulfilling inclusion criteria. No where in the flow chart mentioned about the exclusion criteria.

Page 5, line 166. Why (s. Figure 2)? Why not (Figure 2)?

Page 5, Table 1. Please use dot instead of comma for the mean age, SD, and age.

Page 6, line 178. Please use dot instead of comma for Kendall’s tau and p-value.

Page 6, line 181. Why (s. Figure 3)? Why not (Figure 3)?

Page 7, line 197. Please use dot instead of comma for Kendall’s tau and p-value.

Page 8, line 223. Where is Table 10?

Page 8, line 224. What is (s. attachment)?

Page 8, line 235. What is welche? German word?

Moderate English editing is needed

Author Response

Comments and Suggestions for Authors

This is a systematic review and meta-analysis of cognitive effects of rTMS in Caucasian patients with mild cognitive impairment.

There are few comments here:

Page 2, line 69. The present study conducted a systematic literature search on rTMS in MCI patients in November 2022. The databases used were Medline PubMed, Pubpsych, and Embase, according….. The authors did not do the electronic search in Web of Science (WOS) database, which include the WOS Core Collection, Current Contents Connect, Derwent Innovations Index, KCI Korean Journal Database, Medline, Russian Science Citation Index, and SciELO Citation Index, which covers wider area. The other bigger database is Scopus. Scopus is larger than Web of Science and has more than 23,000 indexed journals in all scientific fields as studied by Falagas ME, Pitsouni EI, Malietzis GA, Pappas G. Comparison of PubMed, Scopus, Web of science, and Google scholar: strengths and weaknesses. FASEB J. 2008;22(2):338–42). If the authors searched in the WOS or Scopus, the outcome might be different (not only 7 studies as mentioned in page 3, line 139).

Thank you for your constructive and detailed feedback. We decided on Embase (Emtree feature) instead of scopus to hopefully provide additional insights from its structured full-text indexing.

Page 2, line 84 to line 93. The exclusion criteria are the negative of the inclusion criteria. Redundant.

Thank you for your feedback. We have deleted the redundant exclusion criteria.

Page 2, line 93. Titles and abstracts, followed by full texts, were reviewed by authors, and hits were scored according to… Should start in new paragraph as the sentences are not part of Exclusion Criteria.

Thank you for your feedback. The sentence starts now in a new paragraph.

Page 3, line 142. Why (s. Figure 1)? Why not (Figure 1)?

Thank you for your feedback, this typo was corrected.

Page 4, Figure 1. 126 articles were excluded because of not fulfilling inclusion criteria. No where in the flow chart mentioned about the exclusion criteria.

Thank you for your feedback. We have deleted the redundant exclusion criteria.

Page 5, line 166. Why (s. Figure 2)? Why not (Figure 2)?

Thank you for your feedback, this typo was corrected.

Page 5, Table 1. Please use dot instead of comma for the mean age, SD, and age.

Thank you for your feedback, was corrected.

Page 6, line 178. Please use dot instead of comma for Kendall’s tau and p-value.

Thank you for your feedback, was corrected.

Page 6, line 181. Why (s. Figure 3)? Why not (Figure 3)?

Thank you for your feedback, this typo was corrected.

Page 7, line 197. Please use dot instead of comma for Kendall’s tau and p-value.

Thank you for your feedback, was corrected.

Page 8, line 223. Where is Table 10?

Thank you for your feedback, was inserted.

Page 8, line 224. What is (s. attachment)?

Thank you for your feedback, was corrected.

Page 8, line 235. What is welche? German word?

Thank you for your feedback, this typo was corrected.

Comments on the Quality of English Language

Moderate English editing is needed

Thank you for your feedback, we will use the language service and correct the whole manuscript as soon as possible.

Submission Date

27 July 2023

Date of this review

07 Aug 2023 10:31:01

Round 2

Reviewer 1 Report

The manuscript does not effectively communicate a clear research question or hypothesis, leaving readers uncertain about the study's purpose and direction.

The manuscript fails to clearly establish the novelty and significance of the research in comparison to existing literature.

Overall the English language is fine.

Author Response

Thank you for your feedback. We are happy that we could improve our English.

We are very sorry that our manuscript does not meet your expectations. We take your feedback seriously and will improve the text.

Reviewer 4 Report

The Authors have addressed all of my concerns with the original manuscript. The revised manuscript is ready for publication

Moderate English language editing is needed

Author Response

Thank you.

We will use the journal's editing service.